# Co-clustering for Federated Recommender System

## ABSTRACT

In an era where data privacy and security are attracting increasing attention, Federated Recommender System (FRS) offers a solution that strikes a balance between providing high-quality recommendations and preserving user privacy. However, the presence of statistical heterogeneity in FRS, commonly observed due to personalized decision-making patterns, can pose challenges. To address this issue and maximize the benefit of collaborative filtering (CF) in FRS, it is intuitive to consider clustering clients (users) as well as items into different groups and learning group-specific models. Existing methods either resort to client clustering via user representations—risking privacy leakage—or employ classical clustering strategies on item embeddings or gradients, which we found are plagued by the curse of dimensionality. In this paper, we delve into the inefficiencies of the K-Means method in client grouping, attributing failures due to the high dimensionality as well as data sparsity occurring in FRS, and propose CoFedRec, a novel **Co**-clustering **Fed**erated **Rec**ommendation mechanism, to address clients heterogeneity and enhance the collaborative filtering within the federated framework. Specifically, the server initially formulates an item membership from the client-provided item network. Subsequently, clients are grouped according to a specific item category picked from the item membership during each communication round, resulting in an intelligently aggregated group model. Meanwhile, to comprehensively capture the global inter-relationships among items, we incorporate an additional supervised contrastive learning term based on server-side generated item memberships into the local training phase for each client. Extensive experiments on four datasets with different sizes are provided. The results verify that CoFedRec outperforms state-of-the-art baselines.

## CCS CONCEPTS

• **Information systems** → **Recommender systems**; **Clustering**.

## KEYWORDS

Federated recommendation, Co-clustering, Supervised contrastive learning

**ACM Reference Format:**
. 2018. Co-clustering for Federated Recommender System. In *Proceedings of Make sure to enter the correct conference title from your rights confirmation emai (Conference acronym 'XX).* ACM, New York, NY, USA, 13 pages. https://doi.org/XXXXXXX.XXXXXXX

## 1 INTRODUCTION

With the rapid development of e-commerce and digital services, people have become increasingly digital-centric [35]. They now spend a significant amount of time online, exploring products, content, and services tailored to their interests. Traditional recommender systems (RS) [1, 47] have proven to be indispensable for e-commerce giants and various digital service providers. However, these systems usually operate by consolidating vast amounts of user data centrally, leading to potential privacy concerns. Federated learning (FL) [4, 25, 41] is a method where multiple clients collaboratively train a deep learning model using their local data. This decentralized approach promotes efficient data exchange and ensures that each participant's data remains private, without being exposed to a central authority or other participants. The Federated Recommender System (FRS) [56, 61] is built on this idea.

FRS is a specialized implementation of FL for recommendation tasks. Instead of directly sending user interaction data to a central server, FRS processes the data locally on users' devices and only the essential model updates are sent back to the central server for global aggregation. Unlike other applications of FL [46, 63], where there are fewer clients and each client possesses a large amount of data from multiple individuals (known as cross-silo FL [21]), in FRS, each user acts as a client constituting only one single user's profile (also known as cross-device FL [22]).

There are an increasing number of works [2, 29, 58] exploring solutions for FRS or the adaptation of cross-silo Federated Learning (FL) algorithms for cross-device contexts. A typical approach involves the utilization of FedAvg [41] to generate a global model and then fine-tune the model on the individual client-side [65]. However, this single global aggregation is inherently designed for IID data. In practical scenarios, the data available on each device is generated or produced by the user which is usually non-IID [14], reflecting different preferences or decision habits across users. To model the heterogeneity across the clients (users), there are works [10, 36, 64] that assume the whole population could be partitioned into distinct clusters or groups, characterized by analogous preferences. On the other hand, collaborative filtering (CF) [18, 42] has proven successful in recommender systems whose power is confined in the federated setting where the whole dataset is not available. Therefore, we can also expect an increase in accuracy by finding out the neighbors of users through clustering and then gathering collaborative insights. In this light, learning a group-level model customized for each group can boost the algorithm's adaptability to heterogeneous clients' data and the ability to transfer useful knowledge among clients by factoring in collaborative insights. In this paper, however, we observe that the widely deployed clustering method which groups the clients using a distance function applied to the updates uploaded by clients [34, 44, 49] in general FL is inefficient in the FRS setting since querying neighbors of high quality is nearly impossible when the feature space is sparse.

To address the aforementioned challenges, we propose a co-clustering mechanism CoFedRec for FRS to effectively group clients without accessing their individual profiles. The core insights come

from (i) the heterogeneity across clients in FR as well as (ii) the understanding of CF whose key idea is to predict the target user's unrated item scores using neighborhood ratings. Specifically, we turn to the experiment results to analyze the inherent limitations of the classical clustering strategies and introduce the co-clustering mechanism. In each communication round, the global aggregation is performed as a preliminary step to gather the global item correlated relationship and yield an item membership via the K-Means clustering technique upon the global item representation. The server algorithm then implements the co-clustering by computing similarity scores among clients regarding a concerning item category, which allows a specific item category to cluster users into two distinct groups, the similar group, and the dissimilar group. Within the similar group, users tend to react similarly towards that type of item. Intriguingly, all the clients in the similar group will update their item embedding network with the aggregated group model while those in the dissimilar group will retain their local model waiting for the subsequent communication round. In addition to the group model, the item membership will be distributed to all the clients. Inspired by the theory of Supervised Contrastive Learning (SCL) [23], a local supervised contrastive term is integrated during the local training phase to include the global item correlated relationship to ensure the locally learned item representation contains the global item insights.

Our contributions are summarized as follows:

- We analyze the failure of classical clustering technique K-Means in the federated recommendation setting and propose a novel co-clustering federated recommendation mechanism CoFedRec which groups users based on a specific item category within each communication round and generates an intelligent group model containing the collaborative information from the user's neighbors. Our proposed paradigm is applicable to different backbones.
- We introduce a supervised contrastive term into the local training phase to include the global item relationship across all users. This ensures that our proposed co-clustering mechanism not only tackles data heterogeneity and user collaborative information but also includes globally diverse insights in local training.
- We conduct extensive experiments on four real-world datasets demonstrating our CoFedRec outperforms state-of-the-art baselines in terms of various ranking metrics.

## 2 PRELIMINARY

### 2.1 Problem Statement

We consider a federated recommender system consisting of a central server and multiple distributed clients where each client represents an individual user. We use $U = \{u_1, u_2, \ldots, u_{|U|}\}$ to represent all users and $I = \{i_1, i_2, \ldots, i_{|I|}\}$ to represent all items where $|U|$ and $|I|$ denote the total number of users and items respectively. Each client corresponds to a user, and each client has its own rating vector $[r_{ui}]_{i=1}^m$ which is given by a user $u$ to an item $i$ and $m$ is the number of items that the user $u$ has interacted with. To protect user privacy, only recommendation models, instead of user data, can be exchanged between the server and the user devices. Thus, the goal of federated recommender systems is to collaboratively train a model for each user to predict the rating of this user for each item $i$ without sharing their individual interaction records.

### 2.2 Failure in User Clustering

The prevailing federated recommender systems draw inspiration from FedAvg [41], i.e., sharing the client's individual recommendation model by a global aggregation, and then the individual clients perform the local fine-tuning [9]. However, due to the diverse preferences among different users, data distribution across clients can be uneven, leading to potential imbalances. The global aggregation without taking into account the discrepancy of these user preferences might introduce undesirable noise in the recommendation results. Additionally, Collaborative Filtering (CF) has proven effective in recommendation systems by leveraging the ratings or interactions of neighbor users who have exhibited similar preferences or behaviors to the target user in the past. Thus, it is intuitive to introduce the clustering to group clients and items before the server-side aggregation, potentially bringing out underlying patterns or similarities among them. The typical clustering methods like the K-Means approach [33, 37], works by computing distances between points, which have proven effective in centralized recommendation scenarios [3, 52, 66] while posing challenges in federated recommendation scenarios.

In the realm of federated recommendation, to protect privacy, direct access to user embeddings on the server side becomes restricted. Instead, we must rely on updates provided by each participant client to execute user clustering. If traditional clustering algorithms are used to solve the above problem, matrix-object data need to be transformed. One of the most significant issues encountered is the curse of dimensionality [20, 26], a problem that arises when we attempt to flatten item embedding matrices resulting in high dimensionality. In high-dimensional spaces, the data points become increasingly sparse, making traditional data processing techniques less effective, as the distance between data points grows larger. This sparsity can make clustering algorithms, like K-Means, less effective as points in high-dimensional spaces tend to be almost equidistant to each other, reducing the algorithm's ability to discern distinct clusters.

To illustrate the challenges further, we analyze the results of applying the K-Means method with $k = 2$ and $k = 10$ on the MovieLens-100k dataset for client clustering. The result of $k = 2$ reveals a highly imbalanced clustering outcome, with user counts drastically skewed: one cluster contains only a single user, and the other contains 942 users. On the other hand, when $k = 10$, the situation does not improve significantly. 8 out of the 10 clusters contain just a single user, the figures are attached in Appendix E.2. This phenomenon verifies the conclusion before, K-Means tends to cluster all the points into one single cluster as the distances between data points become more uniform, which underscores the difficulties of applying traditional clustering techniques to high-dimensional data in federated recommendation scenarios.

Therefore, we can conclude that the use of the traditional clustering method does not necessarily lead to good performance on client grouping under the federated recommendation settings.

## 3 PROPOSED METHOD

In this section, we present our proposed CoFedRec , a novel **Co**-clustering **Fed**erated **Rec**ommendation mechanism which groups

 

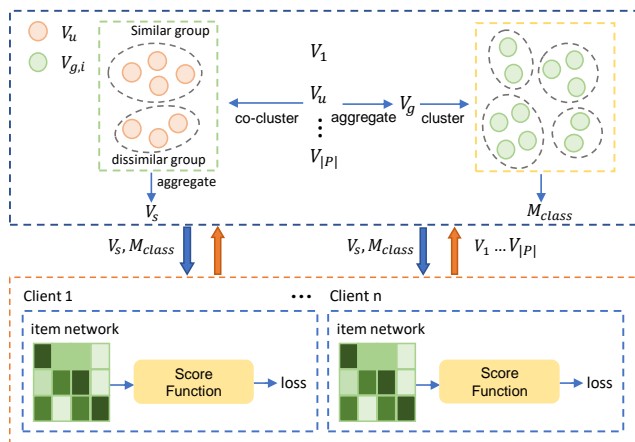

**Figure 1: The overall framework of** CoFedRec **. The pink dots represent the individual models uploaded by participant clients and the green dots are the item embedding vectors of the global aggregation results. Two key parts in** CoFedRec **are (i) Co-clustering mechanism to cluster participant clients into similar group and dissimilar group and an intelligent group model is generated within the similar group; (ii) Supervised contrastive term upon the global item membership is integrated into the loss function of the local training phase to include the global item insights.**

the client upon the item categories on the server side and introduces a supervised contrastive term in the local training phase.

## 3.1 Co-clustering for User Partitioning

We've discussed the importance of user clustering, particularly considering the heterogeneity across clients and the collaborative insights it brings out in Section 2.2. In this section, to address the challenge of user clustering mentioned in the federated recommendation scenario, we propose to generate a group model by aggregating the updates within a potentially similar user group to prevent the transfer of low-quality knowledge.

In light of the observation that it is improbable for users to possess identical interests across all items but rather shared preferences for specific types of items, it becomes natural to classify users based on their affinities for different item categories. Specifically, during each communication round, we focus on a single item category and divide users into two distinct groups based on their likeness or dislikeness for that particular category. To integrate collaborative effects into the learning processes, aggregation is conducted within the user group displaying similar preferences since when users demonstrate shared tastes for a particular category of items, it's probable that they will also have some other common preferences. As such, this aggregation method facilitates the transmission of beneficial insights, incorporating collaborative steps while safeguarding user privacy. Notably, this approach obviates the need to discern whether a congruous group exhibits a predilection or antipathy towards a particular item type.

At each round $t$, the server is required to serve the core client $c \in U$ by presenting an item category $k$, specifically, we find the neighbors of the client $u$ w.r.t item category $k$. To achieve this, the server first performs a global aggregation over all the item network uploads by the participant clients and then generates an item membership $M$ detailing which items fall under which categories. By specifying the item category $k$, we could have all the items belonging to category $k$, denoted as $M_k = \{i | i \in I, M[i] = k\}$. Then we compute the similarity among the core client and all other participants on the selected part of their individual item network. Here we adopt the cosine similarity:

$$s_u = \sum_{i \in M_k} \frac{V_{c,i} \cdot V_{u,i}}{|V_{c,i}| \cdot |V_{u,i}|}, u \in P \tag{1}$$

where $V_{c,i}$ and $V_{u,i}$ represent the vectors of one item under category $k$ of the core client $c$ and participant $u$, respectively. $s_u$ is the similarity score of the the core client $c$ and participant $u$ w.r.t the items category $k$. $P$ is the participant client set. With this equation, we could obtain the similarity score list $S = \{s_1, s_2, ... s_{|P|}\}$.

Upon computing the cosine similarity scores for all participants, to identify the similar group $D_s$ and dissimilar group $D_{dis}$, we propose to use the first elbow point of the similarity scores to divide the participant clients into $D_s$ and $D_{dis}$, which is essentially the point at which the curve starts to level off.

To find the split, we first sort similarity scores as $\{s_{1'}, s_{2'}, \ldots, s_{|P|'}\}$ (here $s_{1'}$ is no longer corresponding to the similarity score of the user 1 but the user with the largest similarity score.) and construct a line that links the first and last points of the similarity scores.

$$\mathbf{L}(x) = s_{1'} + x(s_{|P|'} - s_{1'}) \tag{2}$$

where $x$ is a scalar parameter that determines a point's position along the line $\mathbf{L}(x)$.

For each point $s_u$ in similarity scores, we calculate its orthogonal distance to the line $\mathbf{L}(x)$, which is achieved by projecting $s_u$ onto $\mathbf{L}(x)$ and computing the Euclidean distance between $s_u$ and its projection. Let $h_u = s_u - s_{1'}$ be the vector from the first point $s_{1'}$ to a point $s_u$. The scalar projection of $h_u$ onto $\mathbf{L}(x)$ is given by:

$$x_u = \frac{h_u \cdot (s_{|P|'} - s_{1'})}{|s_{|P|'} - s_{1'}|^2} \tag{3}$$

The orthogonal distance $d_u$ from point $s_u$ to $\mathbf{L}(x)$ can then be computed as:

$$d_u = |h_u - x_u(s_{|P|'} - s_{1'})| \tag{4}$$

the point $e$ with the maximum distance $d_e$ to the line $\mathbf{L}(x)$ is considered the elbow point. This point essentially delineates the optimal neighbors for the core client, denoted as similar group $D_s$, otherwise, dissimilar group $D_{dis}$:

$$\begin{cases} u \in D_s, & \text{if } d_u \geq d_e, \\ u \in D_{dis}, & \text{if } d_u < d_e, \end{cases}$$

Once groups are formed, a group aggregation is performed to transfer the collaborative information based on the common patterns:

$$V_s \leftarrow \frac{1}{|D_s|} \sum_{u \in D_s} V_u \tag{5}$$

All the participants in the similar group will update their item embedding network with the group model $V_s$. With this co-clustering approach, users in the same group might have similar preferences, and thus recommendations can be shared among them more confidently. The clients within the dissimilar group will be disregarded to prevent the transfer of low-quality knowledge among heterogeneous data.

## 3.2 Local Supervised Contrastive Learning

In the previous section, we propose co-clustering to discover common preferences across clients and then cluster them into the similar group and the dissimilar group, excluding the latter during the aggregation phase. However, this could result in ignoring some diverse information, as this global insight might originate from the dissimilar clients that were disregarded.

To consider the global inter-relationships among items, we construct an item membership vector on the server side through item clustering. Recall that at each round, for $N$ participant clients, the server will receive $N$ individual item embedding matrices $V_i \in \mathbb{R}^{|I| \times d}$, $i = \{1, 2, ..., N\}$ uploaded by the participant clients. The server first performs the global aggregation over $N$ local item embedding matrices:

$$V_g \leftarrow \frac{1}{N} \sum_{i=1}^{N} V_i \tag{6}$$

To categorize items, the global item embedding vectors will be grouped into $K$ clusters. We adopt the K-Means to do the item clustering. Assuming that there is $|I|$ items with representation $v_j \in \mathbb{R}^{|u| \times d}$, $j = \{1, 2, ..., |I|\}$ and the number of clusters is $K$. The K-Means method aims to find $K$ centroids $C = \{c_1, ..., c_K\}$, $c_k \in \mathbb{R}^d$, $\forall k \in [K]$ that it uses to define clusters by minimizing the objective:

$$\phi_c(v_j; C) = \|v_j - C\|_F^2 \tag{7}$$

where $\|\cdot\|_F$ denotes the Frobenius norm. Then the set of centroids $C^* = \{c_1^*, ..., c_K^*\}$ gives rise to an optimal segmentation, denoted as $\bigcup_{k=1}^{K} C_k^*$, where $\forall k \in [K]$, $C_k^* = \{v_j : \|v_j - c_k^*\|_F \leq \|v_j - c_m^*\|_F$, $\forall j \in [|I|], m \in [K]\}$.

Upon obtaining the global item cluster results, the server returns an item membership vector $M \in \mathbb{R}^{1 \times |I|}$ to all the participant clients, where the value $M[j]$ at a specific index $j$ indicates the cluster to which the corresponding item $j$ belongs.

Considering that the local client updates the item network based on the server's aggregation, which solely contains the information from its similar neighbor clients after partitioning, it becomes rather restrictive, lacking a comprehensive view that global information can provide. In order to harness the intrinsic similarity and diversity of the items themselves, we incorporate an additional supervised contrastive learning objective into the local training phase.

Supervised Contrastive Learning (SCL) [23] integrates the strengths of both supervised learning and contrastive learning. Utilizing label information, SCL learns representations that bring similar items closer together and push dissimilar items apart, which hence improves the representational quality. In our case, items categorized within the same class are considered as positive pairs, while those from disparate classes are treated as negative pairs. We are able to bring together the representation among items that share similarities, and concurrently, push apart the representation between those belonging to distinct classes by minimizing the following SupContrast term:

$$L_{sup} = - \sum_{i \in I} \log \left\{ \frac{1}{|Z(i)|} \sum_{z \in Z(i)} \left( \frac{\exp(v_i \cdot v_z / \tau)}{\sum_{a \in I \setminus \{i\}} \exp(v_i \cdot v_a / \tau)} \right) \right\} \tag{8}$$

where $Z(i) \equiv \{z \in I \setminus \{i\} : \tilde{y}_z = \tilde{y}_i\}$ is the set of the indices of all positive in the item set. $\tau$ is the temperature parameter to control the uniformity of the representation in the embedding space. $B$ is the batch data during the client's local training.

By leveraging global information to ascertain item membership, our model enhances its ability to recommend items that closely align with users' previous interactions. This thus integrates intricate global inter-relationships present among items within a federated framework.

## 3.3 Overall Workflow

We subsequently develop our federated recommendation via the proposed co-clustering mechanism, detailed in Algorithm 1.

To illustrate the overall workflow of the co-clustering federated recommendation mechanism, we employ a personalized federated recommendation algorithm [65] as our foundational model in the ensuing discussion.

### 3.3.1 Local training.
We first discuss the local training process. In a typical FRS with implicit feedback, each user $u$ has it's rating vector $[r_{ui}]_{i=1}^{|I|}$ where $r_{ui} = 1$ if the user $u$ interacted with item $i$, otherwise, $r_{ui} = 0$. The actual ratings provided by the user are represented by $r_{ui}$, while the predicted ratings are denoted as $\hat{r}_{ui}$.

Each client $u$ holds its own personalized item network $V_u$ and score function $\theta_u$, which is implemented as a one-layer multilayer perception (MLP) here. The client's local dataset $D_u$ is organized as a set of user-item interactions where each interaction is represented as a tuple $(u, i, r_{ui})$. The client's individual item embedding module $V_u$ is updated by the server during each communication round. The objective of the local training on client $u$ is to minimize the binary cross-entropy loss plus the supervised contrastive learning term:

$$L_u(V_u, \theta_u) = - \sum_{(u,i) \in D_u} \log \hat{r}_{ui} - \sum_{(u,i') \in D_u^-} \log(1 - \hat{r}_{ui'}) + \lambda L_{sup} \tag{9}$$

where $\hat{r}_{ui}$ here is computed through the score function $\theta_u$. The client first updates its score function $\theta_u$ using stochastic gradient descent (SGD) and then updates individual item embedding network $V_u$ via SGD as a post-tuning process. $\lambda$ is a hyperparameter to control the linear weight.

### 3.3.2 Server Update.
The server initiates one global model, specifically an item embedding network, used as initial parameters for all client models. During each round, the server begins by randomly selecting a subset of participating clients $P$ and acquiring their item embeddings $V_u$, $u = \{1, 2, .., |P|\}$. Then, the server algorithm randomly identifies a core user and selects an item category based on the item membership $M$ introduced in Section 3.2. Utilizing the user co-clustering technique presented in Section 3.1, the server, for the chosen item category, calculates the similarities between the core user and other participant clients and divides them into the similar

group and the dissimilar group. Within the similar group, aggregation takes place to derive a group-specific model. Subsequently, the group model is distributed back to the corresponding clients.

---

**Algorithm 1** federated recommendation with CoFedRec

**Server Update:**

1: Initialize item embedding $V_0$, item cluster number $K$
2: **for** each round $t = 1, 2, \ldots$ **do**
3:     $P \leftarrow$ (randomly select participant clients for each round from all $N$ clients)
4:     **for** client $i \in P$ **in parallel do**
5:         $V_i \leftarrow$ ClientUpdate$(i, V_s, M)$ $\{V_s = V_0$ for all the clients at round 0$\}$
6:     **end for**
7:     /* Item clustering */
8:     $V_g \leftarrow \frac{1}{|I|} \sum_{i=1}^{|I|} V_i$ {global aggregation}
9:     $M \in R^{1 \times |I|} \leftarrow Kmeans(\{V_{g,i}\}_{i=1}^{|I|})$ {obtaining item membership vector}
10:    /* User partitioning */
11:    $c \leftarrow$ (randomly select a core user for this round from all participant clients $P$)
12:    $[indices_k] \leftarrow$ (randomly select an item category $k$ from item membership and obtain the corresponding indices vector)
13:    **for** client $u \in P$ **do**
14:        $similarity \leftarrow$ SimilaritySocre$(V_u[indices_k], V_c[indices_k])$
15:    **end for**
16:    $D_s, D_{dis} \leftarrow$ (find the elbow point of all the similarities and split the clients into similar group and dissimilar group)
17:    $V_s \leftarrow \frac{1}{|D_s|} \sum_{n \in D_s} V_u$ {aggregating within similar group}
18: **end for**

**CLient Update:**

1: Download item embedding $V_s$ and item membership $M$ from server if the client is in the similar group; Otherwise, only download the item membership $M$
2: Initialize $V_u$ with the latest update
3: Sample negative instances set $D_u^-$ from $\mathcal{I}_u^-$
4: $\mathcal{B} \leftarrow$ (split $D_u \cup D_u^-$ into batches of size $B$)
5: **for** local epoch $e = 1, 2, \ldots$ **do**
6:     **for** batch $b \in \mathcal{B}$ **do**
7:         Compute loss $L_u(V_u, \theta_u)$ with Eq. 9
8:         Model parameters update
9:     **end for**
10: **end for**
11: **Return** $V_u$ to server

---

## 4 EXPERIMENTS

In this section, we conduct experiments to evaluate the performance of our proposed method. Our experiments intend to answer the following research questions:

- **RQ1:** How does CoFedRec perform in the federated recommendation task compared with the baseline models?
- **RQ2:** How do different components in our mechanism contribute to the performance?
- **RQ3:** How good is the generalizability of our proposed CoFedRec ?

- **RQ4:** Do all the clients (users) effectively participate in the cluster-specific aggregation and how good are the clustering results?

### 4.1 Datasets

To evaluate our proposed CoFedRec , we conduct experiments on four datasets with different scales: MovieLens-100K, MovieLens-1M [1] [16], FilmTrust [2] [15], and LastFM-2K [3] [6]. The detailed statistics of each dataset, the preprocessing procedures, and the construction of the training, validation and test sets are shown in Appendix A.1.

### 4.2 Experimental Settings

*4.2.1 Evaluation metrics.* We evaluate the model performance with Top-K evaluation metrics, including Hit Ratio (HR) and Normalized Discounted Cumulative Gain (NDCG). The details of the evaluation metric are presented in Appendix A.2. Following the previous work setting, we fix $K$ as 10 and adopt an efficient sampling strategy that randomly selects 99 items for each user, performing a ranking evaluation among these 100 items.

*4.2.2 Baselines.* We compare our proposed CoFedRec with the following general and state-of-the-art baselines, which contain both centralized and federated methods.
**Centralized methods:**

- **Matrix Factorization (MF)** [27]: Upon the user-item rating matrix, MF maps users and items to a joint latent space, so that the interactions are modeled as the inner product of user and item embeddings.
- **Neural Collaborative Filtering (NCF)** [18]: It proposes to utilize an MLP to model the user-item interaction function.

**Federated methods:**

- **FedMF** [8]: It is a framework implemented based on Federated Collaborative Filtering (FCF) [2] where user embedding is maintained locally and item embeddings are aggregated globally.
- **FedNCF** [45]: It is a federated version of NCF. A generalized MF (GMF) and an MLP are used to represent user embeddings and item embeddings respectively.
- **FedPerGNN** [57]: It assigns GNN models for each client to utilize the superiority of GNN in capturing high-order user-item information.
- **FedRecon** [50]: Utilizing a reconstruction-based approach, FedRecon re-initializes local user embedding every 2 rounds in our implementation and aggregates item network globally.
- **MetaMF** [30]: MetaMF introduces a meta-network to generate private item embedding and rating prediction function so that user model parameters can be reduced. We modify the final layer to adapt to federated recommendations with implicit feedback.
- **PFedRec** [65]: PFedRec proposes a dual personalization mechanism that emphasizes capturing personalized information through a post-tuning procedure.

---

[1] https://grouplens.org/datasets/movielens/
[2] https://guoguibing.github.io/librec/datasets.html
[3] https://grouplens.org/datasets/hetrec-2011/

*4.2.3 Experimental settings.* Following the methodology in [18], we sample four negative instances for every positive instance. Test results are presented based on the optimal validation outcomes. The reported best-performing baseline models are significant w.r.t. the second best performing with p-value < 0.05. Given the inherent variability in our approach, we conducted five runs of our method on each dataset. To provide a conservative estimate, we consistently reported the lowest value from these five iterations. Detailed hyperparameter settings for each dataset across models can be found in the Appendix A.3.

## 4.3 Main Results & Discussion (RQ1)

In this section, we investigate the overall performance of our proposed CoFedRec and the detailed results are shown in Table 1. In three of the four evaluation datasets, namely MovieLens-100K, MovieLens-1M, and FilmTrust, our proposed CoFedRec achieves the best performance even when compared with centralized methods, and on LastFM-2k, CoFedRec still out perform all the federated models.

From Table 1, we derive several insightful findings: (i) When comparing with the strongest baseline method PFedRec, the improvements made by our method are substantial. CoFedRec can obtain a promotion of 1.42% - 9.11% in terms of HR@10 and 2.85% - 15.40% in terms of NDCG@10. A salient one is its performance on MovieLens-100k: 71.05 → 77.52 on HR@10 and 43.89 → 50.65 on NDCG@10. It indicates the superiority of our co-clustering mechanism (including the user co-clustering and local supervised contrastive learning) in exploring the neighbors of both the users and items to enlighten the personal recommendations. (ii) In the experiment, the four datasets we selected are representative, two relatively large datasets and two on a smaller scale. Our proposed CoFedRec surpassed all the federated baseline methods on these four datasets, which proves the robustness of our approach. (iii) If we take a look at the results of the FedPerGNN which lags behind all other baselines, we could draw the conclusion that it is indeed difficult to harness the full potential of GNNs in FRS. Thus, it is compelling to use co-clustering methods to capture collaborative insights and neighbor information. (iv) We note that on some datasets, CoFedRec even outperforms the centralized methods. In fact, centralized recommendations allow for comprehensive user-item interactions to be analyzed globally to generate recommendations. In contrast, FRS faces inherent challenges in capturing nuanced preferences since they often skew the aggregated model towards more common, shared preferences and miss out on more nuanced or niche tastes. The results demonstrate that in federated scenarios, where user-item interactions are processed across scattered local datasets, the co-clustering mechanism can facilitate more insightful aggregation and discover deeper user-user and item-item associations across clients, effectively bridging the gap between centralized and federated approaches.

## 4.4 Abalation Study (RQ2)

In this section, we investigate the effectiveness of each component in our proposed CoFedRec . We denote the state-of-the-art method PFedRec as the original method (Origin in Table 2), which aggregates users without distinguishing users' clusters. We note that our proposed CoFedRec has two main components, server-side client co-clustering and client-side local supervised contrastive learning term. We denote these two parts as User_P and Item_SC in Table 2. For comparison, we also consider client-side similarity learning which is defined as:

$$L_s = -\frac{1}{|D_u|}\frac{1}{|Z(i)|}\sum_{i \in D_u}\sum_{i' \in Z(i)}\left(\frac{i^T i'}{\|i\| \cdot \|i'\|}\right) \quad (10)$$

then the local training loss is replaced with:

$$L_u(V_u, \theta_u) = -\sum_{(u,i) \in D_u}\log \hat{r}_{ui} - \sum_{(u,i') \in D_u^-}\log(1 - \hat{r}_{ui'}) + \lambda L_s \quad (11)$$

From Table 2, we can observe that all the components are very important and designed reasonably. Note that integrating User_P results in a notable performance boost, which verifies the importance of distinguishing similar users and the generation of a group-level model. When we compare the extra local training loss terms, Item_S and Item_SC, the results show that both of these two components have a positive effect on the performance while the latter yields a greater improvement. The primary difference is that the Item_S considers only the alignment between the positive item pairs while the Item_SC focuses solely on aligning positive item pairs, while Item_SC takes into account both alignment and uniformity during local item representation learning. This outcome emphasizes the significance of including the global view information as well. In sum, our co-clustering mechanism, containing the user co-clustering and the local supervised contrastive learning, facilitates the transfer of high-quality knowledge by identifying the effective neighbors while capturing the global item collaborative information.

## 4.5 Geralization Analysis (RQ3)

*4.5.1 Effects on different backbones.* To evaluate the generalization ability of our proposed co-clustering mechanism and local supervised contrastive learning, we compare our methods with the original on three different baselines. The performance is shown in Table 3. Although users' local models are distinguishing, a consistent improvement can be observed when adding our methods. It indicates that our proposed CoFedRec is independent of specific users' local models, and the potential of CoFedRec can be explored extensively with more powerful models. The improvement in MovieLens-100K is more apparent when compared with MovieLens-1M. We attribute it to its smaller quantity of items which enables more faithful item clustering, subsequently leading to more strategic user partitioning and effective aggregation of similarly clustered users.

*4.5.2 Privacy protection with virtual rating.* The primary goal of the FRS is to predict the rating of an item $i$ for a client $u$ without disclosing their rating behaviors or records. Actually, the task of federated recommendation with the implicit data naturally protects user privacy to a certain extent for the reason: (i) it can be seen from the objective function of local training that when using the local dataset to train the model, all items that have not generated actions are treated as negative samples, which indirectly protects the user's behavioral privacy; (ii) we adopted dual personalized proposed by PFedRec as the backbone to preserve the user-specific personalization, meaning that the score function is always kept locally, which prevents the server from inferring the user's behavior

**Table 1: Experimental results on the four real-world datasets through different methods with % omitted. The best results are highlighted in boldface. Underlined values indicate the second best.**

| Models | | MovieLens-100K | | MovieLens-1M | | FilmTrust | | LastFM-2K | |
|---|---|---|---|---|---|---|---|---|---|
| | | HR@10 | NDCG@10 | HR@10 | NDCG@10 | HR@10 | NDCG@10 | HR@10 | NDCG@10 |
| Centralized | MF | 65.43 | 40.16 | 68.61 | 41.33 | 92.09 | 81.99 | 82.88 | 70.81 |
| | NCF | 66.17 | 39.82 | 68.76 | 41.90 | 92.42 | 82.70 | **85.06** | 73.75 |
| Federated | FedMF | 65.11 | 39.13 | 67.52 | 38.12 | 89.49 | 76.31 | 68.44 | 52.97 |
| | FedNCF | 60.13 | 34.31 | 65.78 | 38.67 | 92.34 | 79.87 | 80.19 | 70.11 |
| | FedPerGNN | 35.84 | 19.15 | 43.87 | 24.33 | 92.01 | 82.53 | 72.06 | 57.51 |
| | FedRecon | 65.01 | 38.49 | 60.43 | 34.89 | 91.76 | 81.94 | 82.65 | 67.85 |
| | MetaMF | 66.06 | 39.82 | 45.08 | 25.07 | 92.50 | 82.89 | 81.81 | 66.39 |
| | PFedRec | 71.05 | 43.89 | 73.62 | 44.35 | 91.44 | 82.36 | 82.06 | 73.14 |
| | CoFedRec | **77.52** | **50.65** | **77.75** | **48.81** | **94.05** | **85.20** | 82.75 | **74.10** |

**Table 2: Effectiveness of different components of `CoFedRec` on MovieLens-100K and MovieLens-1M.**

| Models | MovieLens-100K | | MovieLens-1M | |
|---|---|---|---|---|
| | HR@10 | NDCG@10 | HR@10 | NDCG@10 |
| Origin | 71.05 | 43.89 | 73.62 | 44.35 |
| User_P | 75.93 | 47.23 | 73.92 | 45.72 |
| Item_S | 72.75 | 44.26 | 73.66 | 44.67 |
| Item_SC | 73.91 | 44.78 | 74.09 | 44.45 |
| CoFedRec | 77.52 | 50.65 | 77.75 | 48.81 |

**Table 3: Experimental comparisons of `CoFedRec` on MovieLens-100K and MovieLens-1M with different backbones.**

| Models | MovieLens-100K | | MovieLens-1M | |
|---|---|---|---|---|
| | HR@10 | NDCG@10 | HR@10 | NDCG@10 |
| FedMF | 65.11 | 39.13 | 67.52 | 38.12 |
| w/ Ours | 77.09 | 49.90 | 71.39 | 45.10 |
| | ↑ **18.40%** | ↑ **27.52%** | ↑ **5.73%** | ↑ **18.31%** |
| FedNCF | 60.13 | 34.31 | 65.78 | 38.67 |
| w/ Ours | 71.58 | 51.29 | 66.16 | 41.88 |
| | ↑ **19.04%** | ↑ **49.49%** | ↑ **0.58%** | ↑ **8.30%** |
| PFedRec | 71.05 | 43.89 | 73.62 | 44.35 |
| w/ Ours | 77.52 | 50.65 | 77.75 | 48.81 |
| | ↑ **9.11%** | ↑ **15.40%** | ↑ **5.61%** | ↑ **10.06%** |

**Table 4: Performance on MovieLens-100K with varying ratios of virtual rating added to the individual local datasets.**

| Models | Noise size | $\lambda$=0 | $\lambda$=0.1 | $\lambda$=0.2 | $\lambda$=0.3 | $\lambda$=0.4 |
|---|---|---|---|---|---|---|
| PFedRec | HR@10 | 71.05 | 72.00 | 72.96 | 72.53 | 70.52 |
| | NDCG@10 | 43.89 | 44.66 | 44.26 | 44.67 | 43.69 |
| CoFedRec | HR@10 | 77.52 | 75.50 | 75.18 | 75.72 | 72.43 |
| | NDCG@10 | 50.56 | 46.33 | 45.20 | 45.97 | 45.31 |

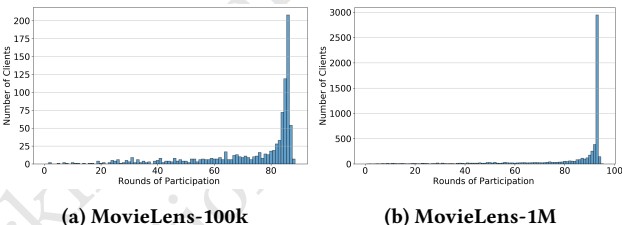

(a) MovieLens-100k    (b) MovieLens-1M

**Figure 2: Distribution of Clients' Participation Rounds on MovieLens-100k and MovieLens-1M Datasets**

through the item network itself. Moreover, FedRec [29] proposes to use virtual scoring during the local training phase together with the true interactions to prevent the leakage of user interaction history when uploading gradients. To rigorously assess the robustness of `CoFedRec`, we employed the virtual rating strategy, sampling items at varying ratios and randomly assigning virtual ratings (either 0 or 1) during local training phases. As shown in Table 4, as the virtual rating ratio increases from 0 to 0.4, `CoFedRec` experiences a slight performance drop and even at a noise ratio of 0.4, our model consistently outperforms all baseline models.

## 4.6 A Close Look at CoFedRec (RQ4)

*4.6.1 Randomness analysis.* In every training round, a user is selected at random to act as the core user. Simultaneously, an item category is randomly chosen to categorize the clients. This inherently introduces randomness into the training process. Consequently, this section examines the involvement of each client in the aggregation of the group model. We evaluated the number of times each client participated in over 100 training rounds on both the MovieLens-100k and MovieLens-1m datasets. The resulting distributions are presented in the Figure 2. From the results, all clients have had an opportunity to contribute to the aggregation of the group model. Specifically, for the MovieLens-100k dataset, 69.64% of the clients participated in more than 70 rounds. In contrast, for the MovieLens-1m dataset, 82.53% of the clients engaged in over 70 rounds. The results highlight that the introduced randomness by our approach does not entirely preclude any client from participating in group

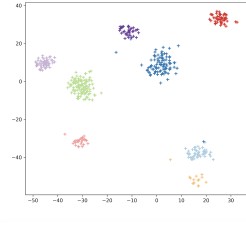
**(a) MovieLens-100K**

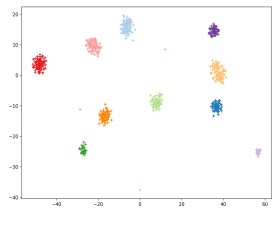
**(b) FilmTrust**

**Figure 3: Visulization of the clustering result on items.**

aggregation. Instead, it facilitates user selection, allowing them to partake in an aggregation process that aligns with their preferences.

*4.6.2 Visualization of clustering effects .* In this section, we analyze the clustering results with our proposed CoFedRec to provide more insight. In Figure 3a, we utilize t-SNE [54] to visualize the item embedding of the global model on MovieLens-100k and FilmTrust (For clear visualization, We randomly select 8 and 10 item categories from the total 30 clusters for two datasets respectively). We can observe that with the extra local supervised contrastive learning, the item embedding is well scattered. Then we consider the quality of the user clustering. Following what we discussed in Section 2.2, in very high-dimensional spaces, even arbitrary data can appear to have structure. While t-SNE is designed to preserve local structures in the data, there's a significant risk of misreading patterns when flattening the individual item embedding matrices into a very high dimension, especially when the number of samples is much smaller than the number of dimensions. Therefore, visualizing the item embedding matrices uploaded by the clients through t-SNE technique may not yield intuitive results The detailed analysis for this can be found in Appendix E.2.

## 5 RELATED WORK

### 5.1 Clustered Federated Learning

**Federated learning (FL)** is a distributed machine learning paradigm which allows a bunch of clients to jointly train a global model without revealing clients' private data to other participants [34, 51, 62]. Based on the participating clients, FL can be classified into cross-device FL [53], involving numerous individual users, and cross-silo FL, which typically considers organizations as clients [19]. Many research efforts in FL address diverse concerns such as communication efficiency [25], privacy [12], data heterogeneity [11], and the cold start problem [55]. **Clustered Federated Learning (CFL)** [5, 13, 60] addresses FL scenarios with diverging, or non-IID, data distributions by clustering similar clients for joint training, mitigating interference from heterogeneous clients. To identify the cluster partitions, Briggs et al. [5] propose a hierarchical clustering step that calculates the similarity of client models to the global model. Sattler et al. [49] introduce a bi-partition method based on the cosine similarity of the client gradients. Mansour et al. [40] assign each client a cluster model that has the minimum loss. Ruan et al. [48] indicate each client can also follow a mixture of multiple distributions and follow this setting to train both local and cluster

models. However, all the above-mentioned methods focus on the cross-silo setting. There's less exploration of the CFL in cross-device settings due to the large quantities and the sparsity of the clients' models. In this paper, we focus on cross-device CFL, especially the problem of federated recommendation (FR), enhancing the FR with the idea of co-clustering.

### 5.2 Fenderated Recommendation

Leveraging the strengths of FL, **Federated Recommendation System (FRS)** offers enhanced recommendations, striking a balance between privacy and decentralized data processing [38]. FCF [2] first applies the thought of collaborative filtering to FRS. This is followed by FedMF [8] and FedNCF [45], which expand upon centralized techniques [18, 27] in the federated context. FedRec [29] studied explicit feedback problems in FRS. FedFast [43] samples participating users in each training round and accelerates the learning to convergence. To enhance personalization, PFedRec [65] retains the score function module locally and integrates a post-tunning procedure. Other techniques like GNNs [32, 39, 57] and meta-learning [30] are also explored to improve the performance of the FRS under various subtopics. Notably, the aforementioned FRS studies produce a singular global model at the server end, potentially introducing noise when users' private data is distributed discrepant. In response to this challenge, there's a shift towards creating group-specific models using clustering techniques, which better cater to diverse user preferences [10, 67]. PerFedRec [36] clusters similar users by user embeddings. FPPDM [31] focuses on multi-domain recommendation, aligning users by their attributes. However, they may risk user profile exposure as user representations are disseminated during clustering. In our research, we develop a co-clustering mechanism that operates on clients' updates, generating an intelligent group model each round while integrating the global insights simultaneously thereby improving the precision and relevance of recommendations.

## 6 CONCLUSION

In this paper, we revisit the significance of clustering in federated recommendation. We analyze the failure of directly applying classical clustering method K-Means in FRS and propose a pioneering **Co**-clustering **Fed**erated **Rec**ommendation mechanism (CoFedRec) for FRS which incorporates two key ideas: (i) To deal with the heterogeneity across clients and harness user collaborative insights, we group clients into similar and dissimilar groups concerning item classifications. This allows for generating a group-specific model tailored to the similar group during each communication round. (ii) Local supervised contrastive learning term is further introduced to include the global correlative information among items. Extensive experiments on 4 real-world datasets demonstrate the superior performance of our proposed method, which outperforms a bundle of baselines. One direction extension of our work is to perform nested client partitioning w.r.t more item categories via our co-clustering mechanism in a single communication round. Moreover, the adaptability of CoFedRec ensures its easy integration with existing FRS. In the future, we'd like to test our model with more advanced backbones.

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

# A EXPERIMENT SETUP

## A.1 Dataset Details

In this section, we introduce the details of the datasets used and how we preprocess the data and construct the training, validation and test set. The two MovieLens datasets record the users' interactions with the MovieLens website over the course of years. Only users who have at least 20 ratings are reserved. FilmTrust is also collected from a movie-sharing and rating website. But interactions in FilmTrust are somewhat less, and accordingly, its sparsity is comparatively high. LastFM-2K contains users' music listening information from the music streaming service Last.fm, where users' listening behavior results in corresponding tags. For FilmTrust and LastFM-2K, we filter out users with less than 5 interactions. The statistics of the four datasets are detailed in Table 5. Given our focus on implicit feedback recommendation in this study, we converted the explicit ratings in each dataset into implicit feedback, specifically, designating a "1" to signify that an item was rated by a user. According to the time stamp of interactions, we employ each user's latest rating record to construct the testing set, the next latest records to constitute the validation set, while all remaining records form the training set.

### Table 5: Dataset Statistics.

| Dataset | Interactions | Users | Items | Sparsity |
|---|---|---|---|---|
| **MovieLens-100K** | 100,000 | 943 | 1,682 | 93.70% |
| **MovieLens-1M** | 1,000,209 | 6,040 | 3,706 | 95.53% |
| **FilmTrust** | 34,888 | 1,227 | 2,059 | 98.62% |
| **LastFM-2K** | 185,650 | 1,600 | 12,454 | 99.07% |

## A.2 Evaluation Protocols

In this section, we give a detailed definition of the two used evaluation metrics, HR and NDCG. Specifically, they are used in a Top-K evaluation setting, where they measure the recommended items in the Top-K list [28, 59]. In the recommended list of length K, if a user-preferred item appears, it is deemed a hit [17]. HR calculates the proportion of hit items in the ground-truth item set. Furthermore, NDCG considers the recommendation ranks and assigns higher importance to the top results. They can be formulated as:

$$HR@K = \frac{1}{N} \sum_{u=1}^{N} hits(u) \tag{12}$$

$$NDCG@K = \frac{1}{N} \sum_{u=1}^{N} \frac{\log 2}{\log(p_u + 1)} \tag{13}$$

Since we select the latest rating records as the testing set and the next latest records as the validation set, there is only one ground-truth item for each user when evaluating. $N$ is the number of users, and it is also the total number of the ground-truth set. $hits(u) = 1$ if user $u$'s ground-truth item in the Top-K list, otherwise $hits(u) = 0$. $p_u$ represents the position of the ground-truth item in the Top-K recommendation list. If it falls outside of the list, $p_u \rightarrow \infty$.

### A.3 Hyperparameter Settings

To ensure a fair comparison across all methods, we maintain a consistent setting: a batch size of 256, an embedding size of 32, and a training round capped at 100. The only exceptions are FedMF, whose convergence needs 300 training rounds, and FedRecon, which does so within 500 rounds. We search for the appropriate learning rate for each model based on the validation sets and the details are shown in Table 6. The hyperparameter $\lambda$ is fine-tuned within the range of [0.0005, 0.001, 0.005, 0.01, 0.05, 0.1, 0.3, 0.5] and the hyperparameter $\tau$ for the local supervised contrastive learning in the range [0.1, 0.5] with the step of 0.1. The specific settings for CoFedRec on four datasets are shown in Table 7. We optimize the centralized MF, NCF, and FedNCF with Adam optimizer [24] and SGD [7] for all the other models.

**Table 6: Learning rate of all models across four datasets.**

| Models | ML-100K | ML-1M | FilmTrust | LastFM-2k |
|---|---|---|---|---|
| MF | 0.001 | 0.001 | 0.001 | 0.001 |
| NCF | 0.001 | 0.001 | 0.001 | 0.001 |
| FedMF | 0.1 | 0.1 | 0.1 | 0.1 |
| FedNCF | 0.05 | 0.05 | 0.05 | 0.05 |
| FedPerGNN | 0.1 | 0.1 | 0.1 | 0.1 |
| FedRecon | 0.1 | 0.1 | 0.1 | 0.1 |
| MetaMF | 0.0001 | 0.0001 | 0.0005 | 0.0001 |
| PFedRec | 0.1 | 0.1 | 0.1 | 0.05 |
| CoFedRec | 0.1 | 0.1 | 0.1 | 0.05 |

**Table 7: Specific settings of our proposed CoFedRec on four datasets.**

| Setting | ML-100K | ML-1M | FilmTrust | LastFM-2k |
|---|---|---|---|---|
| $\lambda$ | 0.005 | 0.005 | 0.05 | 0.001 |
| $\tau$ | 0.1 | 0.5 | 0.5 | 0.5 |
| # item cluster | 30 | 45 | 30 | 500 |
| best round | 93 | 70 | 78 | 68 |

## B COMMUNICATION EFFICIENT ANALYSIS

In this section, we systematically evaluate the communication efficiency of our proposed CoFedRec . Due to the inherent characteristics of federated learning, multiple iterations of parameter exchanges are necessary between the server and the clients to finalize the training procedure. Hence, the efficiency of communication plays a pivotal role in FRS implementations.

To elucidate the superior communication efficiency of our proposed CoFedRec , we'll dissect it step by step. At each round t, a subset of participant clients $P_t$ is chosen to participate. Each client sends its updated item embedding $V_i$ to the server. Thus, the communication cost for item embeddings from all the clients would be proportional to the size of $P_t$ multiplied by the size of each item network. During the client update phase, the server sends the item embedding $V_s$ to the subset of participant clients identified as the

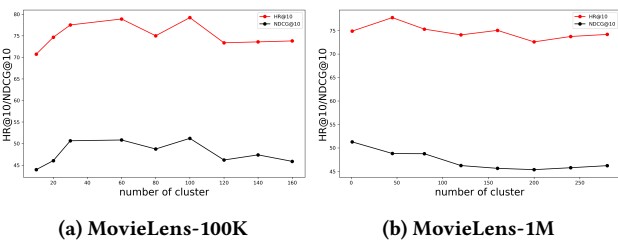

(a) MovieLens-100K  (b) MovieLens-1M

**Figure 4: Effect of the number of the item clusters.**

"similar group" and the item membership $A$ to all the participants. Depending on the fraction of clients in the similar group, this could be a varying portion of $P_t$. The communication cost here is the sum of the sizes of $V_s$ and $A$ multiplied by the number of clients in the similar group where $A$ is a vector in the real implementation and is quite small in size compared with the model $V_s$. Hence, when compared with other baseline models, our proposed CoFedRec stands out as more time-efficient. This efficiency stems from the fact that we eliminate the need to distribute the aggregated model back to all the participating clients.

## C STUDY OF THE HYPERPARAMETERS

In this section, we study the two main factors of our methods, the effect of the number of item clusters and the effect of the local supervised contrastive learning term.

In our co-clustering mechanism, users are grouped based on item categories. Consequently, the quality of item classification directly affects user partitioning. Drawing from practical experience, we consider that after partitioning items, each category should, on average, contain no fewer than ten items. Depending on the size of the dataset, we therefore search for the optimal number of item clusters within the respective range. As illustrated in Figure 4, we plot the performance variations of our proposed method on the MovieLens-100K and MovieLens-1M datasets as the number of item clusters changes. It can be observed that performance declines when the number of item clusters is either too many or too few, due to either over-segmentation or overly broad classification.

We adjusted the weight of the local supervised contrastive learning term from 0.0005 to 0.5 to examine its impact on model performance. The outcomes on MovieLens-100K and MovieLens-1M are presented in Table 8. We found that, when incorporating these extra learning terms with an appropriate weight, they can enhance local item representation learning by capturing more global information.

## D RESULTS ON FULL RANK EVALUATION

In the main experiment, we adopt an efficient sampling strategy. It samples 100 items per user for evaluation, which contain a positive item and 99 randomly selected negative items. In this section, we discard the sampling strategy and evaluate CoFedRec in the full ranking list. It is more challenging because the involved items increase dramatically. We evaluate CoFedRec with baselines on MovieLens-100K, MovieLens-1M and FilmTrust. LastFM-2K is cast away here because its item numbers are too large, and accordingly, it affects the training efficiency. The experimental results are shown in

**Table 8: Performance of varying weights of the local supervised contrastive learning term.**

| Datasets | $\lambda$ | 0.0005 | 0.001 | 0.005 | 0.01 | 0.05 | 0.1 | 0.3 | 0.5 |
|---|---|---|---|---|---|---|---|---|---|
| **ML-100K** | HR@10 | 71.47 | 72.85 | 77.52 | 75.08 | 74.87 | 74.02 | 66.17 | 68.93 |
| | NDCG@10 | 45.47 | 45.43 | 50.56 | 48.86 | 49.01 | 47.36 | 41.05 | 43.36 |
| **ML-1M** | HR@10 | 73.74 | 72.93 | 77.75 | 74.02 | 72.25 | 62.68 | 56.90 | 49.90 |
| | NDCG@10 | 46.09 | 45.60 | 48.81 | 45.66 | 46.15 | 37.11 | 33.41 | 28.77 |

**Table 9: Experimental results on full rank evaluation. The best results are highlighted in boldface. Underlined values indicate the second best.**

| Models | | MovieLens-100K | | MovieLens-1M | | FilmTrust | |
|---|---|---|---|---|---|---|---|
| | | HR@10 | NDCG@10 | HR@10 | NDCG@10 | HR@10 | NDCG@10 |
| **Centralized** | **MF** | 16.76 | 9.26 | 8.84 | 4.47 | 69.44 | 50.35 |
| | **NCF** | 18.98 | 11.56 | 9.67 | 4.89 | 68.87 | 49.00 |
| **Federated** | **FedMF** | 14.10 | 7.16 | 6.61 | 3.14 | 59.66 | 37.10 |
| | **FedNCF** | 15.91 | 8.18 | 7.81 | 3.91 | 48.17 | 35.57 |
| | **FedPerGNN** | 5.73 | 3.15 | 4.21 | 2.16 | 68.95 | 47.31 |
| | **FedRecon** | 16.44 | 8.40 | 8.51 | 4.16 | 68.87 | 49.08 |
| | **MetaMF** | 16.33 | 9.52 | 9.00 | 6.97 | 70.09 | 49.91 |
| | **PFedRec** | 19.19 | 11.02 | 10.13 | 5.04 | 71.37 | 51.82 |
| | **CoFedRec** | **21.63** | **12.64** | **13.20** | **8.90** | **72.78** | **52.93** |

Table 9. It can be seen that our proposed CoFedRec still outperforms all baselines, which indicates its efficacy.

# E  VISULIZATION OF THE CLUSTERING ON MORE DATASETS

## E.1  Item Clustering

In this section, we present the outcomes of our item clustering analysis based on MovieLens-1M. As shown in Figure 5, it is evident that item representations become distinctly spread out, becoming more meaningful by identifying different categories. Such noticeable dispersion can be attributed to our employment of local supervised contrastive learning which significantly enhances the quality of the overall learned item representations.

## E.2  User Partitioning

In our endeavor to understand the problem of K-Means clustering on the MovieLens-100K dataset, as mentioned in Section 2.2, we applied the K-Means algorithm with two different cluster counts: $k = 2$ and $k = 10$. Figure 6 showcases the extremely imbalanced clustering outcome. For example, in Figure 6a, green stars denote cluster 1, while the solitary orange star, representing cluster 2, is nestled within cluster 1. This observation reaffirms our insights from Section 2.2 about the inherent challenges of K-Means in high-dimensional spaces, where data points tend to exhibit near-equidistant relationships, rendering them challenging to distinguish effectively.

We next turn our attention to evaluating the quality of user clustering. As discussed in Section 2.2, traditional data processing methods often underperform when dealing with high-dimensional

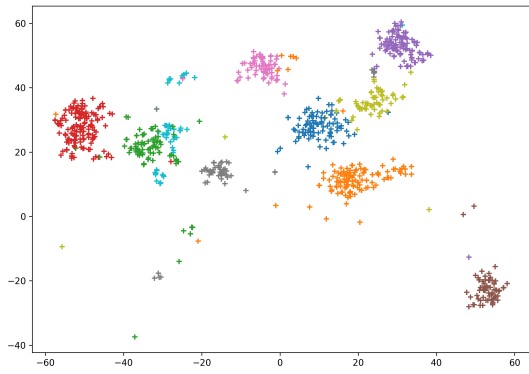

**Figure 5: Visualization of the items clustering results on MovieLens-1M.**

data. This issue becomes particularly pronounced in our settings, where the dimensionality of each sample is considerably larger than the total number of available samples. High dimensionality with a relatively small number of data points can lead to noise in the data and potential overfitting. Specifically, high-dimensional spaces, due to their inherent vastness, can often be deceptive. Imagine having only a few points scattered in an immense space. Even if these points were placed randomly, it might seem like they form some sort of pattern or structure simply because there are so many possibilities for them to potentially align in certain ways. This noise,

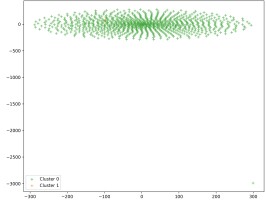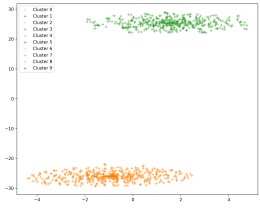

(a) Number of Clusters = 2    (b) Number of Clusters = 10

Figure 6: Visulization for clustering results on MovieLens-100K via K-Means.

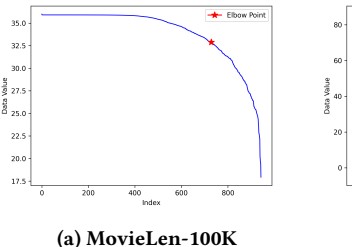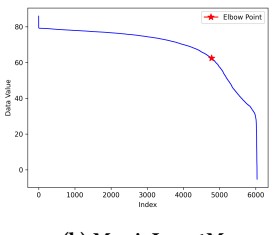

(a) MovieLen-100K    (b) MovieLen-1M

Figure 7: Visualization of the elbow point corresponding to the optimal performance round for both MovieLens-100K and MovieLens-1M datasets.

when interpreted as genuine data structure, can cause models or techniques, like t-SNE, to overfit.

Examining the 'elbow point' used to segregate the similar and dissimilar groups offers further insights. As shown in Figure 7, an evident turning point exists, facilitating the clear differentiation between these groups. This observation underscores the efficacy of our co-clustering mechanism in user partitioning, notably without necessitating access to individual profiles.

