# OpenReview forum: "Co-clustering for Federated Recommender System"
_ACM.org/TheWebConf/2024/Conference — TheWebConf24 Oral_

### Official Review · Reviewer_GFFU · 2023-11-12

**Novelty:** 4
**Technical Quality:** 4

**Review:**

This paper presents CoFedRec, a novel federated recommender system designed to tackle the issue of statistical heterogeneity in Federated Recommender Systems (FRS). CoFedRec addresses this challenge by clustering users and updating models in corresponding clusters. Specifically, in each round, CoFedRec categorizes items into M classes and randomly selects one class to partition users into similarity and dissimilarity groups. Subsequently, users aggregate embeddings only within their respective similarity groups. To enhance model performance, a contrastive learning method is employed. The effectiveness of the proposed method is demonstrated through experiments conducted on four datasets.

The overall motivation of this paper is compelling and promising; however, certain technical details are perplexing. The ensuing discussion delineates the weaknesses and strengths.

Strengths:

S1. This paper investigates a crucial problem in Federated Recommender Systems (FRS): statistical heterogeneity.

S2. The fundamental concept of clustering users and generating personalized models for each user group is straightforward and sensible.

S3. The experimental validation encompasses four datasets, demonstrating the generalizability of the proposed method.

Weaknesses:

W1. In lines 200-210, the authors assert that K-means fails due to the flattening of item embeddings causing a high-dimensional problem. However, an alternative intuitive method for handling matrix-object data is to average or sum them, offering a reasonable approach to obtaining synthetic user embeddings without generating high-dimensional vectors. With these synthetic user embeddings, the clustering and other personalized methods mentioned in this paper could still be achieved. Therefore, the necessity of "co-clustering" is not clear.

W2. The process of obtaining item membership M is unclear in lines 290-300. According to the pseudo-code, is it generated by executing K-means on item embeddings?

W3. Could randomly selecting core clients lead to unexpected results? Especially for MovieLens-100K and MovieLens-1M, which have several "abnormal" clients. For instance, some clients have interacted with nearly half of the items, while others have interacted with only a few (20 items, given the application of 20-core postprocessing). For the former clients, if they are core clients, all remaining clients may be classified into the "similarity" group, while for the latter, most clients will likely be viewed as "dissimilar."

**Questions:**

Please address the concerns in Weaknesses part, especially for W1.

**Ethics Review Description:**

None.

**Reviewer Confidence:**

4: The reviewer is certain that the evaluation is correct and very familiar with the relevant literature

**Scope:**

4: The work is relevant to the Web and to the track, and is of broad interest to the community

---

### Official Review · Reviewer_tnKu · 2023-11-22

**Novelty:** 4
**Technical Quality:** 4

**Review:**

To address the statistical heterogeneity in federated recommender systems, this paper proposes a co-clustering based federated recommendation method, named CoFedRec, which introduces to group clients based on specific item categories. Furthermore, an additional supervised contrastive learning strategy is utilized to capture the global inter-relationships among items. Experimental results on four public datasets demonstrate the effectiveness of the proposed method.

Pros:

1.	The motivation of this paper is well introduced, especially the analysis for failure in user clustering.

2.	The proposed method is technical sound, and makes sense to me.

3.	The experimental results demonstrate the effectiveness of the proposed method.

Cons:

1.	Since the proposed method is clustering-based federated recommendation method, it would be beneficial for the authors to compare it with other such methods, like [1,2,3].

2.	The paper would be benefit from discussing the differences and similarities with related work such as SemiDFEGL [3], which also employs co-clustering techniques in federated recommender systems.

3.	The clustering operation generally is time-consuming, the empirically analysis about model efficiency is missing.

Reference:

[1] Sichun Luo, Yuanzhang Xiao, and Linqi Song. 2022. Personalized federated recommendation via joint representation learning, user clustering, and model adaptation. In Proceedings of the 31st ACM International Conference on Information & Knowledge Management. 4289–4293.

[2] Khalil Muhammad, QinqinWang, Diarmuid O’Reilly-Morgan, Elias Tragos, Barry Smyth, Neil Hurley, James Geraci, and Aonghus Lawlor. 2020. FedFast: Going Beyond Average for Faster Training of Federated Recommender Systems. In Proceedings of the 26th ACM SIGKDD International Conference on Knowledge Discovery & Data Mining (Virtual Event, CA, USA) (KDD ’20). Association for Computing Machinery, New York, NY, USA, 1234–1242. https://doi.org/10.1145/ 3394486.3403176

[3] Liang Qu, Ningzhi Tang, Ruiqi Zheng, Quoc Viet Hung Nguyen, Zi Huang, Yuhui Shi, and Hongzhi Yin. 2023. Semi-decentralized Federated Ego Graph Learning for Recommendation. In Proceedings of the ACM Web Conference 2023 (WWW '23). Association for Computing Machinery, New York, NY, USA, 339–348. https://doi.org/10.1145/3543507.3583337

**Questions:**

1.	How does the proposed method perform compared with other clustering-based federated recommendation methods?
2.	What is the difference between the proposed method compared with the missing related work SemiDFEGL?
3.	How about the model efficiency, such as the convergence efficiency?

**Reviewer Confidence:**

4: The reviewer is certain that the evaluation is correct and very familiar with the relevant literature

**Scope:**

4: The work is relevant to the Web and to the track, and is of broad interest to the community

---

### Official Review · Reviewer_4kWH · 2023-11-23

**Novelty:** 5
**Technical Quality:** 4

**Review:**

This paper presents a clustering-based federated recommendation model. To be specific, it clusters the clients into two groups according to a specific item category and the server only aggregates the local models within the group with similar preferences so that the irrelevant information can be filtered. Besides, by integrating the contrastive loss into local model’s training, the client can absorb the global correlative information.

Pros:
1. The proposed method achieves outstanding performance gain compared with both centralized and federated recommendation baselines.

Cons:
1. There lack of clustered federated learning-based baselines in the experimental evaluation. The paper claims that the current clustered federated learning methods are inefficient in federated recommendation setting. However, the baselines are general centralized and federated recommendation models without clustering-based models, which cannot support the key claim well.
2. The user clustering and item clustering bring significant computation cost. Communication efficiency is a key metric in the federated learning studies and the paper lacks the discussion about the computation cost of the proposed method.
3. Some related works of federated recommendation study are missing, e.g., [1][2].

[1] Qu L, Tang N, Zheng R, et al. Semi-decentralized Federated Ego Graph Learning for Recommendation. WWW, 2023.

[2] Zhang C, Long G, Zhou T, et al. Graph-guided Personalization for Federated Recommendation. arXiv preprint arXiv:2305.07866, 2023.

**Questions:**

1. Why does the proposed method divide the users only based on one item category but not more item categories?
2. What is the time complexity of the proposed method? In each communication round, the server needs to perform cluster for users and items which bring non-negligible computation cost, the authors should give the detailed analysis and show the running time comparison against baselines.
3. Why does the proposed method outperform the centralized baselines significantly in some cases? The authors only provide the conclusion but not convincing analysis in section experiment.

**Ethics Review Description:**

N.A.

**Reviewer Confidence:**

4: The reviewer is certain that the evaluation is correct and very familiar with the relevant literature

**Scope:**

3: The work is somewhat relevant to the Web and to the track, and is of narrow interest to a sub-community

---

### Official Review · Reviewer_eVRy · 2023-11-27

**Novelty:** 5
**Technical Quality:** 6

**Review:**

This paper proposes a novel federated recommendation approach called "CoFedRec". Aiming to address the heterogeneity in client data, this paper proposes to use co-clustering to group clients (users) based on their preferences toward item categories. The main contributions are: 1) It analyzes the challenges of applying traditional clustering methods like K-Means for client grouping in federated recommendation, and proposes a co-clustering approach to effectively partition clients. 2) It introduces a local supervised contrastive learning term during training to incorporate global item relationships. Extensive experiments and analysis on four datasets with different scales demonstrate the effectiveness of the proposed method.
1) The identified limitations of traditional clustering methods for client grouping in the federated recommendation are insightful. The analysis of the curse of dimensionality issue is thorough.
3) Incorporating global item relationships through local supervised contrastive learning is useful to enrich local models. The ablation study proves its contribution.
4) Extensive experiments on four datasets of varying scales demonstrate the effectiveness and generalization ability of the proposed method.

**Questions:**

1) There are some typos, such as:
(1) “…Centrailized method…”
(2)“…CoFedRec still out perform…”
2) While item clustering quality directly impacts user partitioning, how to determine the optimal number of clusters lacks guidelines. More analysis would be helpful.
3) It is suggested to compare with fresh works on federated recommendation, such as [1].
[1] Zhang, Chunxu, et al. "Graph-guided Personalization for Federated Recommendation." arXiv preprint arXiv:2305.07866 (2023).

**Reviewer Confidence:**

3: The reviewer is confident but not certain that the evaluation is correct

**Scope:**

4: The work is relevant to the Web and to the track, and is of broad interest to the community

---

### Official Review · Reviewer_VDpw · 2023-12-01

**Novelty:** 5
**Technical Quality:** 5

**Review:**

The paper discusses CoFedRec, a federated recommendation system that aims to enhance personalized recommendations while preserving user privacy. It introduces a novel co-clustering mechanism that partitions users based on their preferences and aggregates similar user models for more accurate recommendations. The system utilizes local supervised contrastive learning to refine the recommendation model by leveraging both global aggregation and local model updates.
From a technical perspective, the proposed approach provides the following contributions to the research in this field:
- Co-clustering for User Partitioning: The paper proposes a method to aggregate user data based on shared preferences, which helps form user clusters that are then used for personalized recommendations.
- Local Supervised Contrastive Learning: The approach enhances the recommendation system by finding shared preferences across different users and then using this information to improve the clustering of item categories.
- Global and Local Model Updates: It incorporates both global model aggregation and local model updates, leveraging the collective insights from all users while fine-tuning individual recommendations.
- Robustness and Generalization: The paper claims robust performance across multiple datasets, indicating that CoFedRec is not overly dependent on specific data distributions or sizes.

*Pros*:
- Improved Recommendation Quality: By considering user preferences and item categories, CoFedRec can potentially outperform standard federated learning models.
- Privacy Preservation: The system is designed with privacy in mind, using the technical contributions listed above.
- Scalable and Robust: The paper suggests that the system is scalable and performs consistently across various datasets.
- Advanced Learning Techniques: It integrates advanced ML techniques like contrastive learning to improve the model's performance.

*Cons*:
- Presentation:*A few critical concepts in the evaluation and proposed system are not properly discussed (see detailed comments below), making it hard to assess the effectiveness of the method
- Evaluation: The evaluation step needs to be improved in terms of understanding the effect of underlying data and data characteristics, the ablation study, as well as the backbone study.
- Potential Overgeneralization: There might be a risk that the model might overgeneralize user preferences, which could lead to less personalized recommendations.

Comments:
The most important question is why the proposed method performs better results than a centralized setting. There is a general understanding the federated method has inferior performance compared to centralized approaches due to the inherent privacy-accuracy trade-off dilemma. This should be explained clearly and motivated.
The concept of *virtual rating* is defined for the first time in the experimental validation section 4.5.2. It is proper to do this earlier if rating prediction is somehow the key to the FRS superiority.

The authors mention in Section 4.5.1:  “The improvement in MovieLens-100K is more apparent when compared with MovieLens-
1M. We attribute it to its smaller quantity of items, which enables more faithful item clustering, subsequently leading to more strategic user partitioning and effective aggregation of similarly clustered users.”
This could also be linked to the skewness of data distribution. Have you considered providing more insights into this, maybe by considering underlying dataset characteristics? (distribution of rating, size of the catalog, sparsity, among others). The authors could report only the ones that make more sense.
See, e.g.,
- Adomavicius, G., & Zhang, J. (2012). Impact of data characteristics on recommender systems performance. ACM Transactions on Management Information Systems (TMIS), 3(1), 1-17.
- Deldjoo, Y., Di Noia, T., Di Sciascio, E., & Merra, F. A. (2020, July). How dataset characteristics affect the robustness of collaborative recommendation models.
In Proceedings of the 43rd international ACM SIGIR conference on research and development in information retrieval (pp. 951-960).

The variables defined in the 4.4 ablation study are not properly defined: i.e., “User_P”, “Item_C”, “Item_SC”. Thus, it is not very clear which performance is attributed to which component.

In Section 4.3 (RQ1), the authors mention that the graph-based models are inferior in a federated setting, signaling that FRS is not able to capture the potential of these models fully. Could you argue why or why not your model is not able to be used in a graph-based setting and make an argument on the downside and good side of FRS?

Consider adding FedeRank as one of the earliest baselines.

**Questions:**

Since the proposed system relies on the notion of rating, could you clarify if there is any difference in using the proposed method in implicit and explicit settings?
Have you considered providing more insights related to the relation between results and other dataset characteristics, such as distribution of rating, size of the catalog, and sparsity, among others?
How do you propose to solve the overgeneralization issue?

**Ethics Review Description:**

No issue

**Reviewer Confidence:**

4: The reviewer is certain that the evaluation is correct and very familiar with the relevant literature

**Scope:**

3: The work is somewhat relevant to the Web and to the track, and is of narrow interest to a sub-community

---

### Decision · Program_Chairs · 2024-01-22

**Decision:**

Accept (Oral)

**Comment:**

This paper presents a clustering-based federated recommendation model. The reviewers agree that the proposed model is novel. During the discussion phase, the two main concerns, regarding comparing to more recent and appropriate related work, as well as analyzing the model's complexity, have been thoroughly addressed by the authors, who have provided several new experimental results further supporting their claims. Assuming that these will indeed make it in the camera ready version of the paper, I recommend acceptance of this manuscript.